# PLC Cybersecurity Test Platform Establishment and Cyberattack Practice †

**Ramiro Ramirez [1], Chun-Kai Chang [2] and Shu-Hao Liang [2],***

[1]  Department of Industrial Engineering and Management, National Taipei University of Technology, Taipei City 106344, Taiwan
[2]  Graduate Institute of Intelligent Manufacturing Tech, National Taiwan University of Science and Technology, Taipei City 106335, Taiwan
*   Correspondence: shuhaoliang@mail.ntust.edu.tw; Tel.: +886-2-2733-3141 (ext. 5207)
†   This paper is an extended version of our paper published in IEEE.

**Abstract:** Programming logic controllers (PLCs) are vital components for conveyors in production lines, and the sensors and actuators controlled underneath the PLCs represent critical points in the manufacturing process. Attacks targeting the exploitation of PLC vulnerabilities have been on the rise recently. In this study, a PLC test platform aims to analyze the vulnerabilities of a typical industrial setup and perform cyberattack exercises to review the system cybersecurity challenges. The PLC test platform is a sorting machine consisting of an automatic conveyor belt, two Mitsubishi FX5U-32M PLCs, and accessories for material sorting, and Modbus is the selected protocol for data communication. The O.S. on the attacker is Kali ver. 2022.3, runs Nmap and Metasploit to exploit the target Modbus registers. On the other hand, the target host runs the O.S., Ubuntu 22.04 in the cyberattack exercises. The selected attack method for this study is packet reply which can halt operations sending custom data packets to the PLC. In summary, this study provides a basic step-by-step offensive strategy targeting register modification, and the testbed represents a typical industrial environment and its vulnerabilities against cyberattacks with common open-source tools.

**Keywords:** PLC; cybersecurity; automation; industrial Ethernet; communication network; Metasploit; packet reply

## 1. Introduction

The security of critical infrastructure industries is a common element of government defense agencies. Industrial sectors, such as energy, finance, food, transportation, information and communication technology, health, water, or manufacturing, are classified as critical infrastructures according to the Canadian federal government [1]. The incapacity or destruction of critical infrastructure systems would cause negative impacts on cybersecurity, national economic security, and national public health or safety, according to the U.S. Patriot Act of 2001 [2]. The National Institute of Standards and Technology Cybersecurity Framework (NSIT-CSF) provides a tier list to benchmark the cybersecurity progress of a system. The four NIST implementation tiers are as follows; Partial (Tier 1), risk-informed (Tier 2), Repeatable (Tier 3), and Adaptive (Tier 4) [3].

To understand the impact of cyberattacks on industrial companies, Trend Micro, in collaboration with Vanson Bourne (U.K.), conducted an online survey during the first quarter of 2022. There were 900 respondents in total from the United States (300), Germany (300), and Japan (300) who provided their insights in this survey. The participants were part of three primary industries; manufacturing (314), electricity (310), and oil and gas (276). Eighty-nine percent of the participants reported a disruption in their supply chain from cybercriminals per the report by Trend Micro from 2021 to 2022; the duration of this disruption varied across industries. Manufacturing companies reported a disruption of five days, whereas electricity

and oil companies reported a more prolonged disruption, with an average of six days. The economic effects of cyberattacks were considerable for the participants. Manufacturing industries reported average financial damage of USD 1.8 M. Unfortunately, electricity, oil, and gas companies reported higher losses of USD 3.3 M. The status of cybersecurity progress between information technology (I.T.) and operational technology (O.T.) systems was significant. According to the report provided by Trend Micro, 30 percent of the O.T. systems were at the lowest level of cybersecurity (NSIT-CSF Tier-1).

In contrast, 40 percent of I.T. systems recognize cybersecurity risk at the organizational level (NSIT-CSF Tier-2). To hedge this risk, adopting new technologies, such as cloud services or private 5G networks, represent a typical driver for cybersecurity implementation across these industries. In addition, implementing new regulations provides another driver for implementing cybersecurity techniques during the next three years (2023 to 2025) [4].

The status of cybersecurity also varies widely across different nations. The International Telecommunication Union (ITU) helps identify improvement areas regarding national cybersecurity measures. The latest release, the Global Cybersecurity Index of 2020 (GCI 2020), contains extensive information regarding five improvement points: legal measures, technical measures, organizational capacity development, and cooperative measures from 169 ITU member states. According to the ITU, the United States, the United Kingdom, and Saudi Arabia are the current front-runners in cybersecurity worldwide. Regarding cybersecurity progress in the Asia-Pacific region, South Korea, Singapore, Malaysia, and Japan are the regional leaders. In addition, GCI 2020 highlighted the urgent need to increase cybersecurity investment in the industrial sector. According to the ITU, the industrial sector's investment lags compared to similar sectors, such as defense, financial services, or information and communication technologies (ICT) [5].

The increasing geopolitical instability of 2022 led to a surging number of attacks by advanced persistent threat (APT) actors. This situation represents an opportunity window for cyberattackers, which aim to exploit the vulnerabilities of industrial systems. Cyberattacks against industrial conglomerates using O.T. systems have been frequent in this regard. The attacks can be coordinated either by nation-states or state-sponsored groups. The use of cyberattack techniques against critical infrastructure operations from APT actors has been defined as cyberwarfare [6].

Kaspersky Lab summarized the most significant incidents in industrial cybersecurity during 2022. In this regard, hacktivist groups targeted Seliatimo Agrohub (Russia) and the Belarusian railway system to respond to the ongoing war in Ukraine. On the other hand, NVIDIA and Foxconn, significant companies in the electronics industrial space, suffered ransomware attacks. In addition, APT actors targeted Viasat Inc (USA), affecting the service of high-speed satellite broadband for European customers [7].

The cybersecurity research field has an increasing number of contributions regarding PLC vulnerabilities, and many researchers provide extensive literature regarding equipment analysis from manufacturers such as Siemens, Schneider Electric, and Omron. In contrast, the literature regarding vulnerabilities from manufacturers such as Mitsubishi or Panasonic is less common in the cybersecurity community. In addition, industrial companies are reluctant to improve their security after cyberattacks. In this regard, 48 percent of organizations do not take action to reduce future disruptions, according to the survey report provided by Trend Micro. In other words, industrial companies lack security measures, such as intrusion detection systems (IDS) or intrusion prevention systems (IPS).

Based on the previous considerations, our study provides the needed components to simulate a small-scale production line. Implementing a physical test platform provides system modularity allowing the addition or exchange of parts in the industrial environment. The control of the components of the conveyor belt and logic control of sorting relies on two Mitsubishi PLCs (FX5U-32M). This work demonstrates a step-by-step packet reply to halt the operation, aiming to modify the Modbus registers in attack trials.

## 2. Related Works

Critical infrastructure sectors such as electric power plants, transmission grids, or manufacturing facilities require dedicated industrial control systems (ICS). To automate the data collection process, industrial users commonly use systems such as supervisory control and data acquisition (SCADA) and programmable logic controllers (PLC).

To understand the basic concepts of cybersecurity, Nitul Dutta et al. analyzed threats, hardware vulnerabilities, and protective measures against cyber I.T. systems attacks [8]. Integrating SCADA and PLC systems provides flexibility and remote connection capabilities, which are highly valuable for industrial users. However, attacks targeting O.T. systems are common among industrial cyberattacks. Industrial endpoints such as PLC, human–machine interface (HMI), edge devices, or outside computers are frequent targets. In addition, attacks exploiting SCADA vulnerabilities can be traced back to the Cold War [9]. In addition, new state-of-the-art measures, such as two-factor authentication with biometric features, are gaining momentum across the cybersecurity space [10].

The National Security Agency (NSA), in cooperation with the Cybersecurity and Infrastructure Security Agency (CISA), published a whitepaper regarding the main tools used against ICS/SCADA devices in 2022. The report highlighted the development of custom-made tools developed by multiple APT actors. Particular attention should be given to operations using equipment such as Schneider Electric PLC, OMRON Sysmac NEX PLC, and Open Platform Communications Unified Architecture servers (OPC UA) [11].

Acknowledging the inherent risk of SCADA systems, Chen-Ching Liu et al. proposed a vulnerability assessment framework to evaluate the vulnerability level of cyber systems deployed in a power infrastructure [12]. To validate the input of the SCADA system, the research conducted by Gregory et al. suggested the implementation of a "Trust System" architecture for SCADA networks—the proposed system aimed to identify risk and harmful data across an industrial communications network [13].

Phan Duy Anh and Truong Dinh Chau proposed a component-oriented architecture of SCADA software. The study aimed to increase the flexibility and interoperability of SCADA systems [14]. To increase the security of distributed control systems (DCS), Sergi et al. proposed the incorporation of public key infrastructure (PKI), helping to generate a "security by default" implementation scenario [15]. Additional efforts to improve the infrastructure of industrial automation and control systems (IACS) were proposed by Pramod et al. [16]. Rezai et al. summarized the existing key management schemes in SCADA networks, highlighting issues requiring extensive research, such as cryptographic authority or protocol vulnerability assessments [17]. To perform less intrusive assessment analysis on SCADA systems, the research conducted by Adam Hahn and Manimaran Govindarasu highlights the inherent difference between I.T. and O.T. assessment tools. The authors recommend using dedicated tools for O.T. systems, such as security content automation protocol (SCAP) or bandolier [18].

Regarding communication capabilities, the emerging use of private 5G networks represents an opportunity to increase the bandwidth capabilities of large-scale DCS deployments, according to the research of Zhi Lu et al. [19]. However, implementing security standards in industrial control systems is still ongoing. According to William et al., the presence of standards is still relatively low compared to the number of guidelines from government, industry, and standardization bodies [20].

The literature on SCADA protocols, incidents, threats, and tactics is extensive. To provide a curated summary of SCADA systems, Dimitrios Pliatsios et al. conducted a survey where the authors observed current and future trends in this topic. In this regard, the authors highlighted the use of virtualization technologies to reduce the deployment cost of this type of system. In addition, this work also provided a list of threats and tactics to mitigate the vulnerabilities of SCADA systems. Regarding the list of incidents, the team highlighted the fifteen most essential incidents regarding SCADA systems since 2000. These attacks targeted critical infrastructure facilities of countries such as the United States, Saudi Arabia, and Ukraine.

The cyberattack methods range from user compromise to social engineering, viruses, or worms [21]. Regarding future challenges for SCADA networks, Sagarika Gosh et al. highlighted the exposure of existing security standards to quantum computing attacks. Traditional cryptography standards such as Advanced Encryption Standard (AES), elliptic-curve cryptography (ECC), and secure hash algorithms are exposed against Shor's algorithm. The deployment of Shor's algorithm on a quantum computer could decrypt ECC targets. Currently, Bitcoin and Ethereum rely on the use of ECC for transaction verification [22]. Attacks on PLC networks can be executed with a collection of multiple open-source software tools, as the study by Asem et al. suggested [23]. Different attack methods, such as "man in the middle" (MITM) or replay attacks, are commonly seen in PLC network attacks. For this reason, improving PLC systems is a primary concern among researchers and corporations. Hajda et al. highlighted the need to improve the security standards regarding industrial communication protocols, network architecture, integrity checking, and software updates and reduce the human component element [24].

The adoption of Modbus as the selected data communication protocol comes from its extended adoption in industrial environments. The research conducted by Gonzales et al. used a smart grid physical testbed to collect and represent the data obtained by multiple actuators. The data communication between sensors and the supervisory system was possible using the Modbus protocol [25]. In addition, Modbus can be combined with other protocols to increase its flexibility. Due to its industrial nature, Modbus is a synchronous protocol. Complementary protocols such as message queuing telemetry transport (MQTT) can be considered to achieve asynchronous communication. Samer Jaloudi compared multiple communication protocols regarding industrial Internet of Things applications. The combination of Modbus and MQTT provides flexibility to the whole system. However, using multiple protocols increases industrial data's payload [26].

Regarding Modbus security, Santiago et al. suggested a central architecture for the validation phase. This "validation" concept has evolved into sandboxing technologies, providing virtual environments to confine the actions of a specific malware into an isolated environment. The authors proposed a new secure version of the Modbus protocol, where a central architecture design authorizes the client, server, and its Modbus frame [27]. The current version of Modbus (Modbus TCP Security) guarantees the integrity and confidentiality of the established session. Modbus TCP Security uses port 802 as a safe port for communication, whereas the previous versions used port 502. In addition, public key infrastructure (PKI) and role-based access control (RBAC) are also included in its latest release. Martins et al. combined Modbus TCP Security with a MITM component to validate both the client and server [28]. The measures mentioned above are defensive, aiming to increase the security of SCADA systems and the Modbus protocol.

In contrast, offensive measures aim to exploit the vulnerabilities of industrial hardware and software. Industrial application devices, such as PLC or controller modules, could expose an industrial operation to third-party attacks. Rongkuan et al. exposed the vulnerabilities of UWNTEK equipment in their research study. The team found a vulnerability in the UW5101 controller module, where the reboot command (0114H) did not require authentication and authorization when executed. This vulnerability provides access to the root user and allows telnet commands during restart, creating a backdoor in the firmware code. This backdoor allows complete control and execution of critical commands using remote access [29].

Henry et al. extensively analyzed the vulnerabilities of Siemens PLC devices. The team generated valid network packages to conduct a replay attack, targeting Siemens S7 devices (S7-1211 PLC). The attacks comply with three main parts: packet capture (pcap), reverse engineering, and session attack. Firstly, the team performed a packet capture using Wireshark, an open-source packet analyzer. Secondly, the team conducted a reverse engineering analysis of two Siemens Firmware versions (v4.1 and v4.2). This analysis found firmware vulnerabilities which helped to craft an appropriate packet response. The study used WinDbg, a multipurpose Microsoft Windows operating system debugger, to

perform the exercise. After analyzing the main attributes of the message, the researchers used Scapy, a packet manipulation tool for computer networks written in Python. Finally, using a custom Scapy script [30], the team conducted a replay attack with valid network packages that exposed the vulnerabilities of Siemens S7 devices [31].

Dragos Inc., a state-run cybersecurity firm founded by the United States government, recently released information about Pipedream. Pipedream is an industrial control system (ICS)-specific malware developed to disrupt industrial processes. Specifically, Pipedream targets PLCs from Omron and Schneider Electric, possibly targeting and attacking controllers from multiple additional vendors. The released whitepaper lists the leading models vulnerable to malware attacks. In addition, the whitepaper declares that it is possible to target industrial technologies, such as Modbus Transmission Control Protocol (Modbus TCP), OPC UA, Codesys, and Windows. The main objective of this whitepaper is to provide a standard operation procedure (SOP) to avoid cyberattacks from Pipedream or similar malware attacks on both public and private institutions [32].

## 3. Materials and Methods

This section describes the materials and methods used in this study, including a description of the components used for the PLC test platform and a reference of the open system interconnection model (OSI model) protocol stack. In addition, a list of the multiple industrial communication networks and their components presents the comprehensive networks and depicts a brief system network architecture connection. This work is an extension of the previous research conducted by Ramiro et al. [33].

### 3.1. Automatic Conveyor (Test Platform)

Conveyor belts are used in industrial environments for transportation and sorting procedures. This type of automation system relies on PLCs, which control the movement of the motors and the sensor signals. Disruption in the conveyor belt system leads to production delays or even an operational halt of the assembly line for hours, representing substantial economic losses for manufacturers. The selected test platform application was designed with the following list of components, as Table 1 shows.

**Table 1.** Test platform components.

| Model | Test Platform Components | | |
| --- | --- | --- | --- |
| | Reference | Type | Connection Bus |
| Mitsubishi | FX5U-32M | PLC | RS-232, RS-485, GPIO, I2C |
| Wicocc | DC12V 66RPM | Motor | GPIO |
| Tend | TP-SM5N1 | Sensor | GPIO |
| Omron | E3Z-D61 | Sensor | GPIO |
| Weinview | MT8051iP [1] | Display | RS-232, RS485 |

[1] Human–machine interface (HMI) with 4.3″ TFT LCD Display.

Observing industrial cyberattacks' behavior can utilize physical or digital testbed environments. Regarding digital environments, Nitul Dutta et al. highlighted the flexibility of using hypervisors. Hypervisors allow simulation data transmission and cyberattacks in a controlled environment [34]. In addition, using software simulators such as OpenPLC enables testing new PLC functions in virtual environments [35]. Unfortunately, OpenPLC only supports Modbus and Distributed Network Protocol-3 (DNP3) SCADA. The research conducted by Muhammad M. Roomi et al. provided additional support for IEC 61850, releasing the program as an open-source project [36].

In contrast, this study focuses on physically implementing a test platform for cybersecurity research. Figure 1 illustrates the interconnection of the main components in the cybersecurity test platform. The local computer links all the components in the local network via RS485 and Ethernet. WISE-4051 is a WiFi/RS485 converter that can transmit the controller data via WiFi. On the right of Figure 1, the remote computer plays the role of the

attacker in the cyberattack trials. The human–machine interface (HMI) is the manipulating device for the operator, and it can be a cyberattack target too.

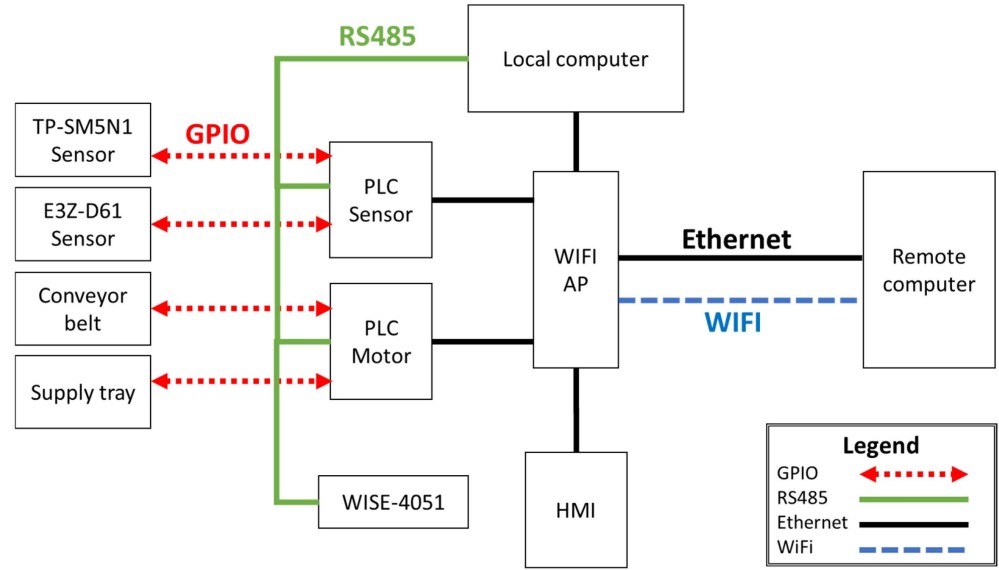

**Figure 1.** Network and components connection of the PLC cybersecurity test platform.

The physical cybersecurity test platform was built for cyberattack trials at the Industry 4.0 Implementation Center, part of the National Taiwan University of Science and Technology (NTUST), as Figure 2 shows. The supply tray feeds working pieces into the conveyor and guides working pieces into the sliders for plastic and metal, respectively. In the case of cyberattacks, the sensors and motors might not be functional at normal status.

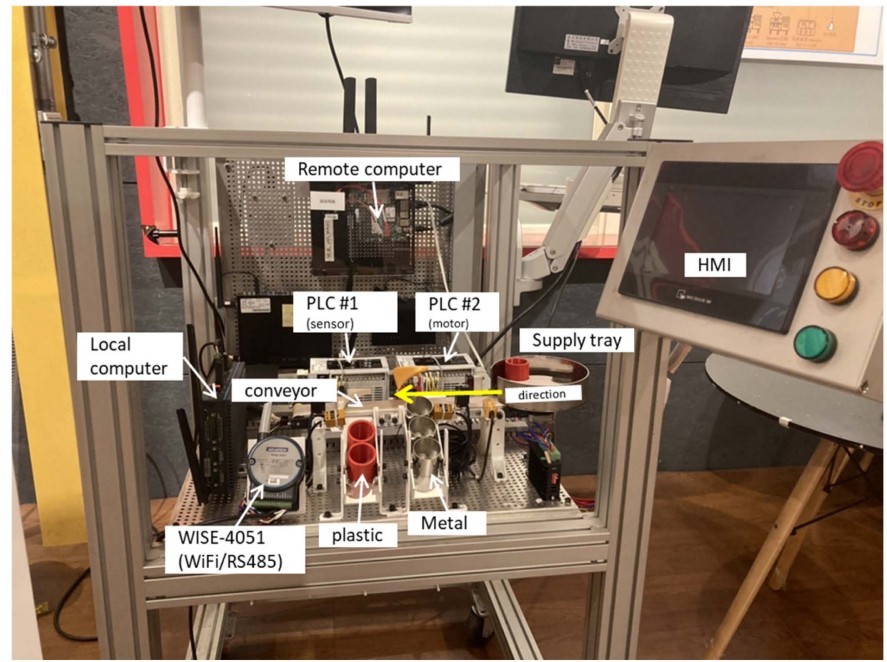

**Figure 2.** PLC cybersecurity test platform.

### 3.2. Heterogeneous Industrial Networks

Using different connections provides enhanced communication possibilities and flexibility to the industrial network. The primary network connection for this study is Ethernet, the ASUS RT-N12HP, as the initial communication setup, which can bridge the other facili-

ties with different protocols, such as WiFi A.P., LPWAN gateway, or 5G adapter for specific requirements. The Advantech WISE-4051, an RS-485/WiFi converter, can connect the PLC and the components only with RS-485 ports. Table 2 shows the initial components of the heterogeneous network in an industrial environment.

**Table 2.** Network Components.

| Brand | Components for Heterogeneous Network | | |
| --- | --- | --- | --- |
| | Model | Type | Connection |
| ASUS | RT-N12HP | Router | Ethernet, WiFi |
| Advantech | WISE-4051 | IoT gateway | RS-485, WiFi |

### 3.3. Protocol Stack

The protocol stack selected for this application is represented in Table 3. Modbus TCP and SLMP are the selected communication protocols to analyze for industrial communication.

**Table 3.** Protocol Stack.

| OSI Layer | Protocol Stack |
| --- | --- |
| Application | Modbus TCP, SLMP |
| Transport | TCP |
| Network | IPv4, IPv6 |
| Datalink | Ethernet, WiFi |

### 3.4. Software Toolset

Penetration testing is an authorized cybersecurity attack simulation to evaluate the target system's security. Currently, there is a vast offering of software dedicated to cybersecurity purposes. In this regard, Kali OS is a Debian-derived Linux distribution designed for penetration testing. Additional tools, such as Nmap, used for network discovery of I.P. addresses and ports [37]; Wireshark, used for packet capture analysis (pcap) [38]; or Metasploit, used for executing predefined scripts to run commands against the target [39], are part of the Kali OS ecosystem [40]. This set of tools is the leading software component for the attacker user.

The target user used Ubuntu 22.04 as the primary operating system, and ModbusPal v1.6 was utilized to emulate the communication protocol used by multiple PLCs. This emulator simulates an industrial communication environment between the host (target) and the attacker. The list of cybersecurity tools used for this study can be found in f Table 4.

**Table 4.** List of cybersecurity tools.

| Software | Cybersecurity Test Tools | | |
| --- | --- | --- | --- |
| | Version | Type | Role |
| Kali | 2022.3 | Penetration O.S. | Attacker |
| Nmap | 7.93 | Network discovery | Attacker |
| Metasploit | 6.1.34 | Exploit code | Attacker |
| Ubuntu | 22.04 | Operating system | Target |
| ModbusPal | 1.6 | Modbus emulator | Target |
| JavaSDK | 18.0.21 | Java Distribution | Target |

### 3.5. System Network Architecture

The network is composed of three primary devices: an Attacker (Kali 2022.3), a Target (Ubuntu 22.04), and a PLC (FX5U-32M). Interconnection of a PLC to the network relies on the ASUS RT-N12HP, which works as the middle point in the equipment interconnection and can be our reference to observe network communication and network security. Figure 3 summarizes the connection between components in the network.

Modbus communication tends to be a critical point in the industrial network. The connection between Modbus and SCADA is critical for the operations of extensive facilities. In this study, the network connection between the target (Ubuntu 22.04) and its ground equipment (FX5U-32M) relies on an Ethernet connection. This study observes access to an authenticated user within the industrial network.

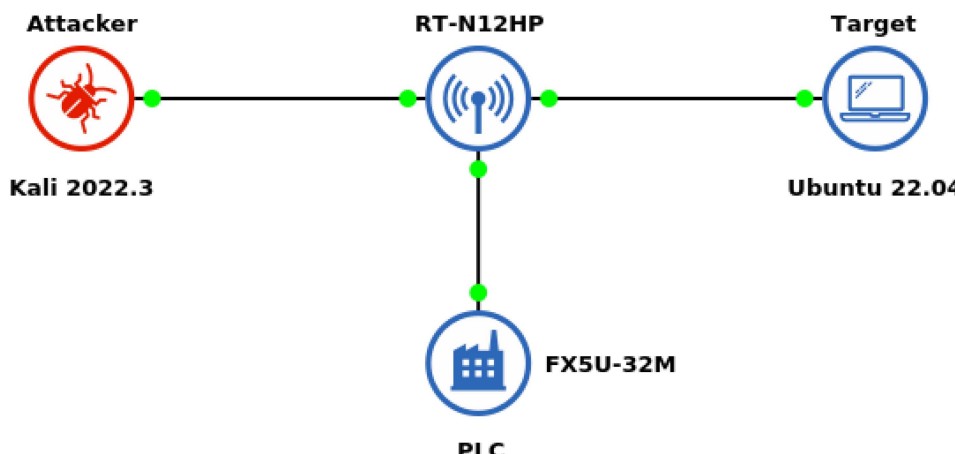

**Figure 3.** System network architecture. (attacker–target connection).

The security of industrial communication networks tends to be underestimated due to the lack of awareness of operational technology (O.T.) ports and protocols, which can significantly differ from information technology (I.T.) standards. This study assumes a lack of security measures for the industrial network. This assumption is based on the report provided by Trend Micro, where 30 percent of the O.T. systems are at the lowest level of cybersecurity (NSIT-CSF Tier-1). In addition, the occurrence of insider attacks is frequent among industrial corporations. The insider can be a member of the organization, an associate (contractor, business partner, or guest), or a person previously associated with the corporation. Due to its relevance in international cybersecurity, the North Atlantic Treaty Organization (NATO) has defined multiple insider detection methods in its Insider Threat Detection Study, published in 2014 [41]. To mitigate insider threats, the Cybersecurity and Infrastructure Security Agency (CISA) provides multiple guidelines in its Insider Threat Mitigation Guide [42].

*3.6. PLC commands*

The selected PLC, Mitsubishi FX5U-32M (Melsec iQ-F Series), controls the movement of the conveyor belt. The model includes an Ethernet connection, referred to in the users' manual as the "Ethernet module." The user manual provides an extensive collection of commands and ports using SMLP for data communication developed by Mitsubishi Electric [43]. The most common port for SLMP communication is 1025. However, using 1024 to 5558 and 5570 to 61,439 is recommended. The list of remote commands to control the PLC module is in Table 5.

**Table 5.** MELSEC IQ-F series remote commands.

| Name | Command | Sub-Command |
| --- | --- | --- |
| Remote Run | 1001H | 0000H [1] |
| Remote Stop | 1002H | 0000H |
| Remote Pause | 1003H | 0000H |
| Read Type Name | 0101H | 0000H |

[1] Ethernet module command.

## 4. Attack Trials

This section explains the cybersecurity tools used in the step-by-step cyberattack approach. Firstly, a personal computer running Kali 2022.3 as its kernel operating system plays the attacker. Then, a personal computer running Ubuntu 22.04 as its kernel operating system works as the target device. For this testing scenario, we assume the attacker has access to the local network via Ethernet to the access point (ASUS RT-N12HP). Kali contains the needed cybersecurity tools to generate a simple reply attack.

The target device runs ModbusPal v1.6 to simulate Modbus communication between the target and the industrial equipment. We assume the attacker does not know the target's I.P. address or port. The following points describe the sequence of actions and the software involved at each stage to disrupt the operation of this test platform environment.

### 4.1. Scanning I.P. and Ports (Nmap)

The first step to disrupt the operation of the test platform is to observe the target's I.P. address and port. To do this, the attacker must execute a network mapping of the network to identify the devices connected. Assuming the attacker has access to the WiFi access point, he can execute Nmap commands from the terminal console of its operating system (Kali S). Modbus uses port 502 for communication, which can be the target for exploitation in attacks. After obtaining the scan report, Nmap provides valuable information, such as MAC, I.P. address, and port status. Figure 4 represents the Nmap scanning procedure.

**Figure 4.** Nmap network scanning (find I.P. address and port status).

### 4.2. Obtain Register Values (Metasploit)

Metasploit contains a collection of scripts that can be executed from the terminal console. To observe the values of the Modbus registers, Metasploit can request the register values of each address. Figure 5 represents the use of Metasploit to obtain the register values of a specific address in Modbus.

```
msf6 auxiliary(scanner/scada/modbusclient) > run
[*] Running module against 192.168.31.129

[*] 192.168.31.129:502 - Sending READ HOLDING REGISTERS...
[+] 192.168.31.129:502 - 1 register values from address 2 :
[+] 192.168.31.129:502 - [78]
[*] Auxiliary module execution completed
```

**Figure 5.** Read holding registers.

*4.3. Modify Modbus Registers (Metasploit)*

The use of Metasploit allows the modification of registers in the selected target. In this case, we decided to modify the register at the second address. To demonstrate the registry modification, we decided to write 4444 as the new registry value. The previous value was 78, as shown in Figure 5. Figure 6 contains the Metasploit write command and its result.

```
msf6 auxiliary(scanner/scada/modbusclient) > run
[*] Running module against 192.168.31.129

[*] 192.168.31.129:502 - Sending WRITE REGISTER...
[+] 192.168.31.129:502 - Value 4444 successfully written at registry address 2
[*] Auxiliary module execution completed
```

**Figure 6.** Metasploit write command (new registry value).

Once the registry has been modified anonymously, the system could be compromised. The new value has been written in the holding registers section of ModbusPal. Figure 7 shows the changes in the red frame, address 3, value 4444.

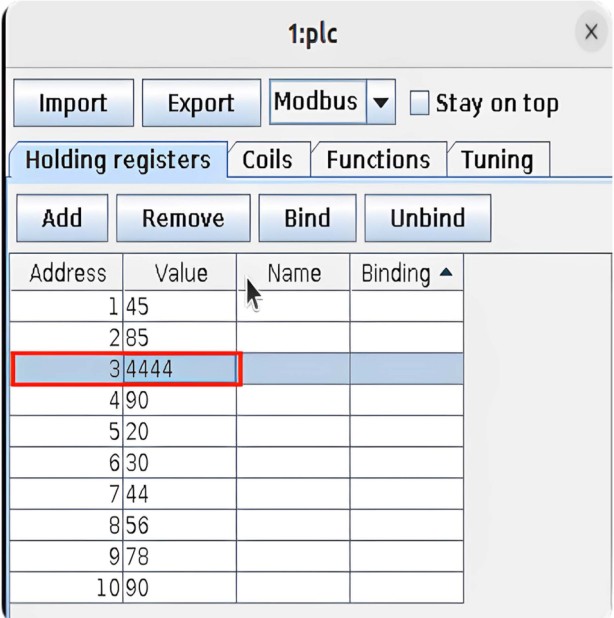

**Figure 7.** Holding registers modification (system compromised).

The penetration software tools (Nmap, Metasploit) have been executed in the attacker environment (Kali 2022.3). Figure 8 summarizes the main steps to achieve this replay attack, and the combination of the previous steps represents the attack workflow of this experiment.

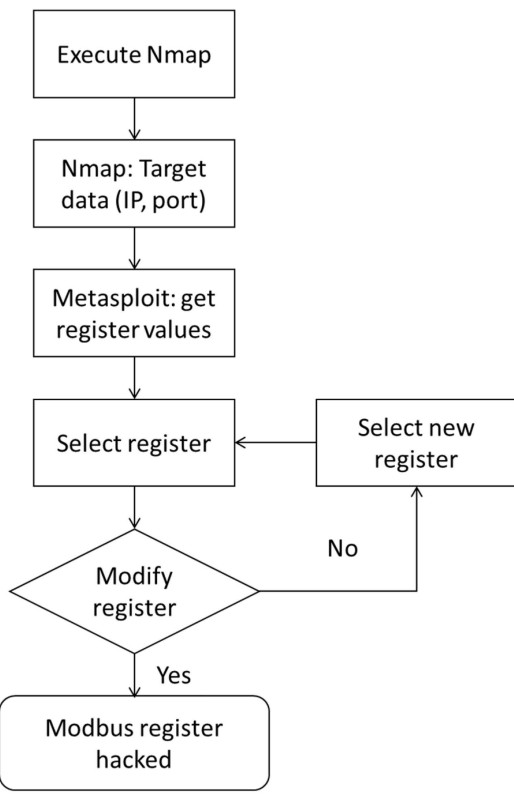

**Figure 8.** Cyberattack workflow (Kali 2022.3).

### 4.4. System Architecture for Training Purposes

Establishing a reliable test platform provides the opportunity to conduct data mining activities, which could also be leveraged for future machine learning and artificial intelligence solutions. The adoption of evolutionary computation techniques in cybersecurity environments reflects the increasing interest in this kind of solution among the research community and industrial corporations [44]. Using Support Vector Machine (SVM) for large amounts of intrusion data has been explored by Prashanth et al. [45].

Our team foresees that sensing abnormal network activities through artificial intelligence can be a significant information security technology development. Figure 9 represents the proposed system architecture for training an artificial intelligence model based on our current test platform.

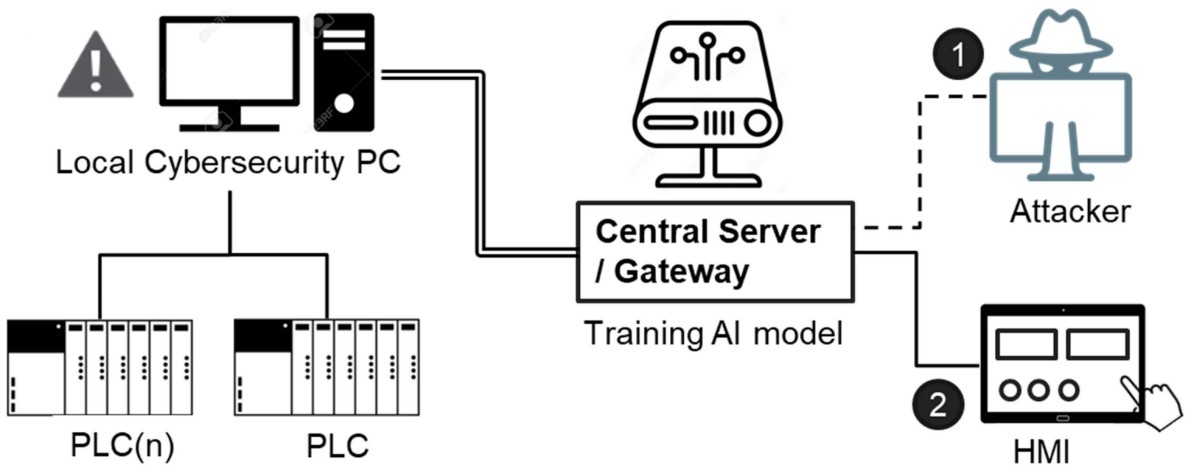

**Figure 9.** Proposed system architecture for training an A.I. model.

## 5. Discussion

Establishing an industrial cybersecurity testbed environment provides a safe space to observe the common vulnerabilities of industrial networks. Ethernet is the primary wired computer networking technology due to its high adoption in industrial Internet environments, where clients prioritize wired connections over wireless solutions. Therefore, the Industrial Internet of Things (IIoT) and Industry 4.0 systems require secure Ethernet connections for data transmission. The use of Mitsubishi FX5U-32M as our selected PLC responds to two primary needs; firstly, it is a widely adopted model of PLC in the Asian region, and secondly, the analysis of vulnerabilities of this type of equipment is moderate compared to leading brands such as Siemens or Omron.

Modbus is a standard data communication protocol in industrial scenarios. However, this study highlights the vulnerabilities and the availability of software tools to disrupt industrial communications. Due to the lack of security measures, such as intrusion detection systems (IDS) or intrusion prevention systems (IPS), the network is vulnerable to attacks. Access to the local router provides communication across the whole network, facilitating packet reply attacks. In this study, the attacker could generate a network mapping and write registers due to the lack of preventive measures. Unfortunately, this lack of security is common among industrial networks, as the 2022 Trend Micro survey mentioned. This work provides information for industries on network cybersecurity tasks through the test platform and cyberattack demonstration.

Industrial operators could adopt open-source solutions with IDS and IPS capabilities to mitigate these risks. In this regard, Snort performs real-time traffic analysis, which allows the detection of stealth port scans, such as the one conducted by Nmap in this study [46]. In addition, due to different industrial needs, the industrial network can adopt various network components. Wired solutions such as RS485 or Ethernet are conventional for Larger Area Networks (LAN). In contrast, wireless solutions, such as WiFi, LPWAN, or 5G, provide higher flexibility and can increase the data transmission distance between client and server topologies. The heterogeneous communication network environment in industrial scenarios requires a more detailed study. In this regard, the upcoming private 5G networks represent an opportunity to increase the security of industrial networks [19].

## 6. Conclusions

The easy access to open-source tools for penetration testing increases the risk and frequency of cyberattacks. Industrial scenarios are especially vulnerable due to a lack of investment, awareness, and the use of legacy industrial protocols. Multiple researchers have proposed central architecture designs to avoid this risk, where device validation is required to establish communication within the industrial network. The replay attack was possible due to the end user's lack of authentication and security systems such as firewalls or trunk ports. This study represents the first step in developing new tools to mitigate cybersecurity attacks in industrial networks containing PLC devices. The presented practical application scenario provides a testing environment for cybersecurity tools. The current experiment was conducted over Ethernet networks, known as Industrial Ethernet, and this type of communication is currently expected in industrial scenarios.

## 7. Future Work

Wireless WiFi, LoRaWAN, or 5G solutions enable network flexibility between client and server. Wireless solutions are attractive due to reduced hardware components, which can be significant in large-scale production lines. Large area networks and wide area networks (LPWAN) are desirable. Network connections relying on WiFi devices, such as the Advantech WISE-4051 (WiFi) and Fibocom FM150 (5G) modules, are currently under research for future studies.

**Author Contributions:** Methodology and hardware, R.R. and C.-K.C.; measurements and data curation, C.-K.C.; and writing—original draft preparation, S.-H.L. All authors have read and agreed to the published version of the manuscript.

**Funding:** This research was funded by NSTC 111-2218-E-011 -016, the funded project of the National Science and Technology Council (NSTC) of Taiwan. The Center for Cyber-Physical System Innovation (CPSi), National Taiwan University of Science and Technology (NTUST), Taiwan, provided the experiment field and equipment support.

**Institutional Review Board Statement:** Not applicable.

**Informed Consent Statement:** Not applicable.

**Data Availability Statement:** The data presented in this study are available on request from the corresponding author. The data are not publicly available due to their use in future research activities.

**Conflicts of Interest:** The authors declare no conflict of interest.

## Abbreviations

The following abbreviations are used in this manuscript:

| | |
|---|---|
| AES | Advanced Encryption System |
| APT | Advanced Persistent Threat |
| DCS | Distributed Control System |
| ECC | Elliptic-Curve Cryptography |
| GPIO | General-Purpose Input/Output |
| HMI | Human-Machine Interface |
| I2C | Inter-Integrated Circuit |
| IACS | Industrial Automation and Control Systems |
| ICT | Information and Communication Technology |
| IDS | Intrusion Detection System |
| I.P. | Internet Protocol |
| IPS | Intrusion Prevention System |
| I.T. | Information Technology |
| LPWAN | Low-Power Wide Area Network |
| MITM | Man-in-the-middle |
| OPC UA | Open Platform Communications Unified Architecture |
| O.S. | Operating System |
| OSI | Open Systems Interconnection |
| PKI | Public Key Infrastructure |
| PLC | Programmable Logic Controller |
| SCADA | Supervisory Control and Data Acquisition |
| SLMP | Seamless Message Protocol |
| TCP | Transmission Control Protocol |

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
