# Peer review of "PLC Cybersecurity Test Platform Establishment and Cyberattack Practiceâ€"

_electronics, doi:10.3390/electronics12051195_

Round 1
Reviewer 1 Report
The manuscript covers an interesting R&D topic and fits the scope of the Journal. PLCs cyber-security is currently a very relevant field of research given their importance in modern Industry 4.0 facilities and requires new advancements and proposals. The paper requires some extra efforts to improve its quality and presentation. A set of comments are expounded hereafter.
- Regarding the format of the document:
Abbreviations must be decomposed the first time that they appear within the text. For example, TCP/IP, OPC-UA, are directly used in line 122. Other examples along the manuscript are DNP3 (line 173), OSI (line 155).
On the contrary, in line 243 it is not required repeating Seamless Message Protocol.
In line 160, “PLC logic controllers” is found. This is redundant, so only PLC should be used.
The format of references must be revised to match the template of the Journal.
- About the content of the manuscript, as aforementioned, it covers an interesting topic. The comments after a careful revision are the following:
Additional keywords are proposed to achieve a higher visibility: Communication network, Automation.
In general, the manuscript is too short and should be enlarged with proper content. Along the comments, some suggestions could be useful in this sense. In addition, a brief list or enumeration or scheme of the main security vulnerabilities in PLC-based automation networks could be included in the manuscript. For example, vulnerabilities associated to the PLC (firmware), to the SCADA software (and to the operative system that supports it), network protocols (where the paper puts the focus), and even to basic aspects like user security (password, physical access to devices) or implementation of DMZ and firewalls. This would enhance the perspective of where the reported research is placed as well as to improve the overview of cyber-security in industrial automation systems.
Some comment about the Industry 4.0 paradigm should be included in the introductory section given the fact that PLC are essential for this paradigm and that the testing environment is placed in the Industry 4.0 Implementation Center of the NTUST. Even more, mentioning such paradigm would enhance the visibility of the paper and, hence, the interest of the readers.
A common practice in scientific papers consists on describing briefly the structure of the rest of the manuscript to boost the readability.
The third section, Motivation and contribution, should be placed within the Introduction, from this humble reviewer viewpoint.
The Modbus TCP protocol is chosen as object study. This is a good choice given its wide-spread use in industry as well as the cyber-security weaknesses that have already been reported in literature. In this regard, the authors should mention in an explicit manner the reasons to choose this protocol on the view of previous literature. Some recent publications that could be considered by the authors in this regard are now given. Firstly, the application of Modbus TCP in many types of facilities should be emphasized by the following papers:
- Innovative Multi-Layered Architecture for Heterogeneous Automation and Monitoring Systems: Application Case of a Photovoltaic Smart Microgrid. Sustainability 2021, 13, 2234. https://doi.org/10.3390/su13042234
- Communication Protocols of an Industrial Internet of Things Environment: A Comparative Study. Future Internet 2019, 11, 66. https://doi.org/10.3390/fi11030066
Secondly, the security vulnerabilities that Modbus TCP presents must be contextualized, for example, the Modbus TCP Security version was released in 2020 and is different from the common specification, and for example, it uses the port 802 instead of the 502. This aspect could be pointed through the next paper:
- Enhanced Modbus/TCP Security Protocol: Authentication and Authorization Functions Supported. Sensors 2022, 22, 8024. https://doi.org/10.3390/s22208024
Finally, the following paper also deals with PLC-related security issues and has been left unnoticed:
- Security Challenges in Industry 4.0 PLC Systems. Appl. Sci. 2021, 11, 9785. https://doi.org/10.3390/app11219785
A photograph of the physical equipment (PLC and conveyor belt) would improve the description of the research. Additionally, a block diagram or scheme depicting the common linkage between PLC, SCADA and other equipment in industrial facilities could be drawn and introduced in the fourth section.
The conducted attack trials are very briefly described; only two pages is excessively short for one of the most relevant sections. Therefore, more details should be given.
The flowchart of figure 6 is very illustrative.
The Discussion conducted in the sixth section is to brief. It should be enlarged through three main aspects. The main strengths of the research must be highlighted, indicating the contribution to the field of knowledge. For example, if none of previous literature has used the software tools (Metasploit, etc.) in the same manner, it could be indicated. Another aspect to mention in this section are the main limitations of the work for a comprehensive description of the research. An additional aspect to enrich this section, as well as the whole paper, is commenting, at least a paragraph, the implications of the achieved results in the context of modern industries, such as those Industry 4.0-enabled infrastructures.
5G communication receives a whole paragraph within the Conclusion section. However, the paper does not deal with such communication, so less text should be devoted to 5G in this section. Other option consists on placing the 5G paragraph to the Discussion section, and in the Conclusion mentioning only a future work related to 5G.
Another issue occurs with the title. Namely, the title and the abstract mention explicitly “security challenges”; however, within the text, this term is not found. Even more, some previous paper already contains this term in the title (https://doi.org/10.3390/app11219785). Therefore, this reviewer strongly suggests modifying the title to avoid misleading to the reader and for a proper description of the paper contents.
Author Response
Thanks for your comments. We have responded to your valuable comments in the attachment.

Reviewer 2 Report
Please indicate what is the difference between this paper and your paper published in
R. Ramirez, C. -K. Chang and S. -H. Liang, "PLC Cyber-Security Challenges in Industrial Networks," 2022 18th IEEE/ASME International Conference on Mechatronic and Embedded Systems and Applications (MESA), Taipei, Taiwan, 2022, pp. 1-6, doi: 10.1109/MESA55290.2022.10004463.
this is a special Issue focused on post-conference full journal paper publications for technical and research work presented at the 9th International Conference on Advanced Robotics and Intelligent Systems (ARIS 2021) not 2022 18th IEEE/ASME International Conference on Mechatronic and Embedded Systems and Applications (MESA).
Author Response

(The authors gave the same response as above.)

Reviewer 3 Report
Dear authors,
Your topic is inserting, and you gathered good bunch of information, but there are some points you need to consider; please kindly apply these recommendations:
1.      The title can be changed to more comprehensive one.
2.      The abstract section is not comprehensive, it needs correction and needs more details in your work; add results in that too.
3.      Keywords needs improvement.
4.      In first 5-7 you frequently repeated the information about the scope and the application many times, you could make it simpler and not repeat them again and again.
5.      You need to review other related state of work works; related works are not appropriate.
6.      Your English needs work, it is not precise and scientific, change it to a better one or use grammatical online programs which can help you. Correct all the grammatical and syntax errors and wrong usage of the English language should be revised.
7.      Paper's language has many problems in using verbs with double meaning which is very critical in PLC, and security levels, you used some words which has negative meaning in one area and then use them exactly in other sections with such same words; as an expert or tenderfoot reader of this field, reader can't figure out which meaning do you intended to use either positive or negative.
8.      Pictures' fonts are small in many cases, you can make them more readable.
9.      Some parts and tables like Table 2 is not justified in your paper margin's. they need to be justified and aligned well.
10.     Cyber Security aspects are not covered well.
11.     Algorithms and comparison of them are very vague and missing important parts.
12.     Your paper organization is not good, messy in many parts, and where you should have expanded and explained the paragraphs you got concise and where you should have been concise you got expanded. You are doing the wrong thing right.
13.     What are contributions of the paper?
14.     Where are the other approaches and comparing with them?
15.     The issues and methods for solving these issues are not clear.
16.     You can use diagrams more in your paper, since the paper is short, it gets dull soon. Besides it helps readers to see what you mean.
17.     Diagram for your sections is small and fonts are not suitable
18.     You need a table for "all" abbreviations.
19.     One of the main issues with your paper is you widen your major and for this wide scope you need to comparison cross disciplinary analysis.
20.     Charts and diagrams are small.
21.     Most of your sections are not comprehensive about its subject, for instance Cyber security Topologies is too concise and not even half satisfying.
22.     In most cases you didn't used any kind of reference for your tables or your paragraphs didn't use any kinds of them.
23.     Recheck all of your references, there are some mistakes.
24.     Some of your references are not eligible for using in scientific paper.
25.     Comparisons are not satisfactory and comprehensive, besides state of art works are not completely covered in them.
26.     What is Take Away section! What's for? if you like to make a section for future works, write it instead of this messy way of organization.
27.     Conclusion section is not comprehensive in any aspect which it should be.
28. State of the art articles are mising from the related work section and also include one compariosn table of all state of the art approaches with the propsoed one. Suggested to include following references:
https://link.springer.com/book/10.1007/978-3-319-98734-7
https://link.springer.com/chapter/10.1007/978-981-16-6597-4_1
https://link.springer.com/chapter/10.1007/978-981-16-6597-4_8
Author Response

(The authors gave the same response as above.)

Round 2
Reviewer 1 Report
The new version of the manuscript has properly addressed the reviewers concerns and has been noticeably enhanced. Congratulations to the authors for their efforts.
Author Response
Thank you for your previous comments and suggestions regarding the quality of the manuscript. The current version has improved thanks significantly to your contribution and review process.

Reviewer 2 Report
In this paper, a PLC test platform aims to analyze the vulnerabilities of a typical industrial setup and perform cyberattack exercises to review the system cybersecurity challenges. This study provides the needed components to simulate a small-scale production line.
The paper can be reconsidered after minor revisions. I have the following concerns:
1- It is strongly advised to write paper in own words not copied even from own publications.
2- The figures quality not good especially, Figs. 1, 7 and 8.
3- The limitations of the presented platform should clearly be discussed.

Author Response

(The authors gave the same response as above.)

Reviewer 3 Report
Great work by the authors in revision
No more comments
Author Response

(The authors gave the same response as above.)
